# Combining host-derived biomarkers with patient characteristics improves signature performance in predicting tuberculosis treatment outcomes

Dhanasekaran Sivakumaran [1,2], Synne Jenum [3], Mario Vaz[4], Sumithra Selvam[5], Tom H. M. Ottenhoff[6], Marielle C. Haks[6], Stephanus T. Malherbe[7], T. Mark Doherty [8], Christian Ritz [9✉] & Harleen M. S. Grewal[1,2]

Tuberculosis (TB) is a global health concern. Treatment is prolonged, and patients on anti-TB therapy (ATT) often experience treatment failure for various reasons. There is an urgent need to identify signatures for early detection of failure and initiation of a treatment switch. We investigated how gene biomarkers and/or basic patient characteristics could be used to define signatures for treatment outcomes in Indian adult pulmonary-TB patients treated with standard ATT. Using blood samples at baseline, a 12-gene signature combined with information on gender, previously-diagnosed TB, severe thinness, smoking and alcohol consumption was highly predictive of treatment failure at 6 months. Likewise a 4-protein biomarker signature combined with the same patient characteristics was almost as highly predictive of treatment failure. Combining biomarkers and basic patient characteristics may be useful for predicting and hence identification of treatment failure at an early stage of TB therapy.

[1] Department of Clinical Science, Faculty of Medicine, University of Bergen, Bergen, Norway. [2] Department of Microbiology, Haukeland University Hospital, University of Bergen, Bergen, Norway. [3] Department of Infectious Diseases, Oslo University Hospital, Oslo, Norway. [4] Department of Physiology, St. John's Medical College and Division of Health and Humanities, St. John's Research Institute, Koramangala, Bangalore, India. [5] Division of Infectious Diseases, St. John's Research Institute, Koramangala, Bangalore, India. [6] Department of Infectious Diseases Group, Immunology and Immunogenetics of Bacterial Infectious Disease, Leiden University Medical Center, Leiden, the Netherlands. [7] DST-NRF Centre of Excellence for Biomedical Tuberculosis Research, South African Medical Research Council Centre for Tuberculosis Research, Division of Molecular Biology and Human Genetics, Faculty of Medicine and Health Sciences, Stellenbosch University, Cape Town, South Africa. [8] GlaxoSmithKline Vaccines, Wavre, Belgium. [9] Department of Nutrition, Exercise and Sports, University of Copenhagen, Copenhagen, Denmark. ✉email: ritz@nexs.ku.dk

Standard treatment for pulmonary tuberculosis (PTB) caused by drug-sensitive *Mycobacterium tuberculosis* (*Mtb*), is a 6-month regimen with multiple drugs. Typically treatment involves a 2-month intensive phase (rifampicin, isoniazid, ethambutol and pyrazinamide) followed by a 4-month continuation phase (rifampicin, isoniazid) that provides a bacteriologic cure (treatment success) rate of 83% globally[1]. Effective therapy is crucial to any strategy for controlling TB and the emerging drug-resistant (DR) and multidrug-resistant (MDR) TB epidemic[2]. More efficacious treatment regimens are needed for TB. However, drug development is impeded by a scarcity of reliable biomarkers of disease severity and treatment efficacy. Further, the currently available tests, such as the Gene Xpert® MTB/RIF test, only detect *rpoB* gene-related rifampicin resistance and is far from being a point-of-care test in high-endemic low-resource areas[3]. Also, not all treatment failure is related to drug-resistance. There is a need for prognostic biomarkers of treatment efficacy, preferably enabling identification of patients with high risk of treatment failure even before the start of treatment, for tailored management[3]. There is broad scientific consensus that such future prognostic tools must be based on biosignatures rather that single biomarkers[4], thus reasonable analysis platforms are multiplex assays based on transcriptional or soluble protein biomarkers in peripheral blood. Recently, studies applying RNA profiling in peripheral whole blood (WB) have been used for the unbiased search for biomarkers/biomarker signatures with the capacity to improve TB diagnosis and treatment monitoring[5–8]. However, previous studies have reported changes in biosignatures during treatment rather than evaluating their predictive capacity for defined treatment outcomes. In a well-characterized South-African TB cohort (the Catalysis treatment response cohort, "CTRC"), Thompson et al.[9] evaluated the previously identified 16-gene correlate-of-risk (COR) signature[8], which revealed symmetrical differences in the gene expression signatures between TB progressors and TB patients during treatment. The COR signature also had the capacity to ipredict treatment failure (defined as culture positivity after 4 and 24 weeks of ATT) in their CTRC cohort. By applying a considerably less resource-intensive method than RNA sequencing, the dual-colour-Reverse-Transcriptase-Multiplex-Ligation-dependent-Probe-Amplification (dc-RT MLPA) platform[10] we have explored pre-selected panels of host transcripts in Indian children with PTB and identified with excellent diagnostic and predictive capabilities[11–14].

In the practical management of TB patients, general socio-economic and medical assessments are done as part of the diagnostic process to assess disease extent, infectivity and evaluate treatment compliance. These assessments reveal that TB patients are highly heterogeneous, making it unlikely that the WHO Standard treatment regimen would fit all. Biomarkers of treatment response therefore have an undisputable potential for guiding tailored individual ATT as well as for greatly facilitating clinical trial evaluation for alternative TB treatment regimens/strategies[15–20].

The present study aimed to investigate the predictive capacity of baseline transcriptional and soluble protein biomarkers to predict TB treatment outcomes in an Indian adult TB cohort. However, as most patients are subject to clinical assessment at baseline and also re-assessment after 2 months of ATT when conversion to the continuation phase is anticipated, re-assessing predictive capacity of patients characteristics and biomarkers is high relevant at 2 months as high risk of Treatment Failure could justify diversion from the WHO Standard ATT by prolonged intensive and/or continuation phase or added/altered drugs. The present study aimed to derive signatures at both 2-month and 6-month treatment outcome. A further novelty of the present study is that signatures also take into account patient characteristics at diagnosis that are considered relevant to treatment outcome.

## Results

**Patient characteristics.** For a total of 90 patients out of 164 which were enroled in the cohort study and which provided written informed consent, blood samples were obtained at baseline. Transcriptional biomarkers were measured for 90, 62, 67 and 61 patients at baseline, 1 month, 2 months and 6 months, respectively (see Fig. 1). Protein biomarkers were, however, only measured for 83, 56, 59 and 55 patients at baseline, 1 month, 2 months and 6 months, respectively.

In this study, the sputum culture negative conversion rate after 2 months of treatment is 51.1% (46/90), and the TB Cures and Treatment Failures (based on sputum culture status at 5 month) rates are 61.1% (55/90) and 38.9% (35/90), respectively. The rate of TB cures in the present study is low compared to WHO-reported data from India where treatment success rate (based on smear examination) for all TB cases (new and relapse) was 88% in 2012/2013[1]. However, the mean age was 45 years (range 18–75 and males constituted 83% (75/90) of TB cases, 57% used tobacco or smoked (last 6 months) and 40% had consumed alcohol (last 6 months) (Table 1).

For 84 patients in the external validation data the mean age at diagnosis was 33 years (range: 17–64), males constituted 61% of TB cases, and ~75% were current smokers.

**Prediction of treatment outcome at 6 months.** Baseline signatures for discriminating between TB Cures and Treatment Failures after 6 months ATT derived using either patient characteristics or transcriptional biomarkers only appeared to be very similar with identical AUC values of 0.78 (Table 2). However, the signature combining patient characteristics and transcriptional biomarker (*AIRE, B2M, CD19, CXCL10, CXCL13, NCAM1, NLRC4, NLRP1, NLRP2, NOD2, TLR6* and *TLR8* combined with gender, tobacco, alcohol, previously TB diagnosed and severe thinness) data led to an improved prediction corresponding to an AUC value of 0.89 (Fig. 2a). This picture was even more pronounced for 2-month signatures (the signature combining patient characteristics and transcriptional biomarkers included *BCL2, CXCL10, IL13, STAT2, TLR6* and *TLR8* combined with age, gender, BMI, cough ≥2 weeks, tobacco, alcohol, previously diagnosed TB, and severe thinness) and AUC values were also in general slightly higher, indicating higher predictive capability (Fig. 2b). The corresponding signature coefficients are reported in Table 3: For most biomarkers down-regulation implied a higher chance of being cured. Also, mostly different biomarkers were included in baseline and 2-month signatures, only *CXCL10, TLR6* and *TLR8* overlapped. The relative importance of included patient characteristics and biomarkers in the signatures of combined data are shown in Fig. 3a and b for the baseline and 2-month signatures, respectively. Notably, a number of biomarkers appeared to be as important as the patient characteristics for predicting the treatment outcome (*AIRE, CD19, CXCL10, CXCL13, NCAM1, NLRP1, NOD2, TAP2, TLR6* and *TLR8*; down-regulation meant improved prognosis for all these biomarkers), i.e., biomarkers are not all weakly predictive. Moreover, some biomarkers became more important from baseline to 2 months (*CXCL10* and *TLR6*).

Signatures based on baseline patient characteristics or baseline protein biomarkers alone resulted in comparable AUC values whereas the signature based on combined baseline data (Eotaxin3, GMSCF, IL7 and MDC combined with gender, tobacco, alcohol, previously diagnosed TB and severe thinness) performed better (Table 2, Fig. 2c). The corresponding 2-month

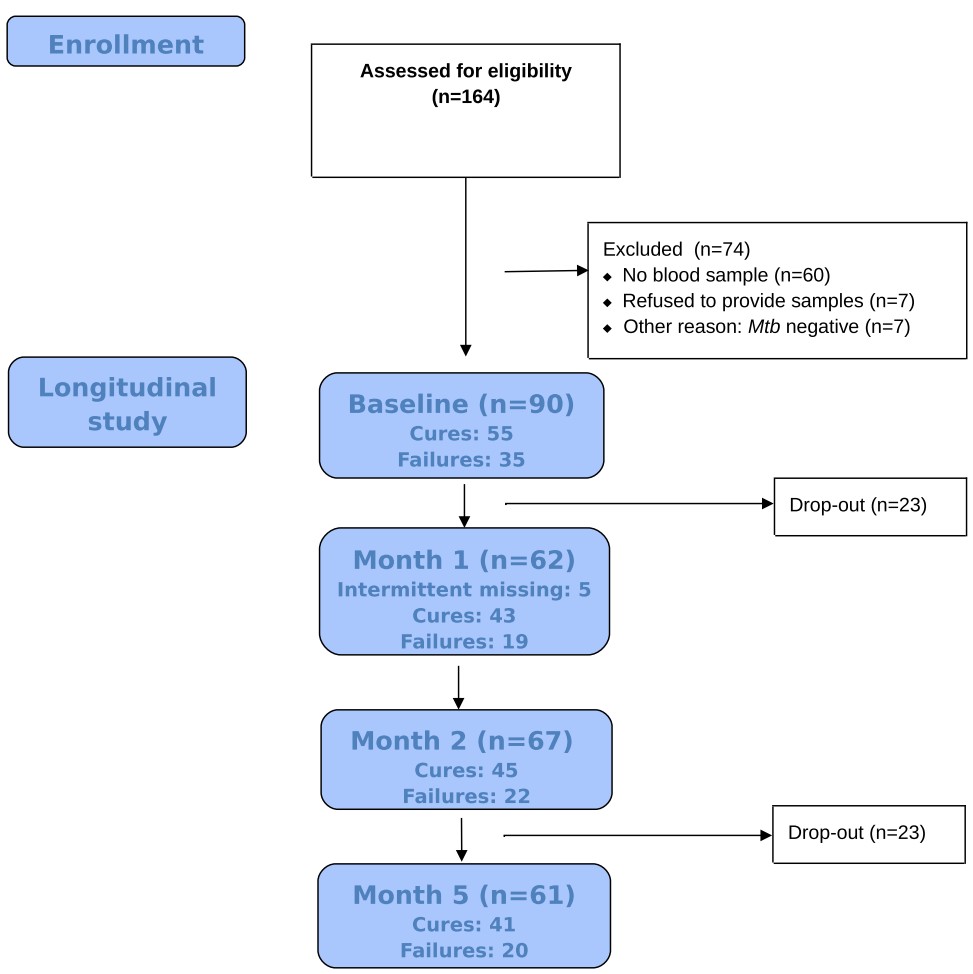

**Fig. 1 Study flow chart.** Mtb *Mycobacterium tuberculosis.*

**Table 1 Patient characteristics (n = 90) shown for all TB cases and separately for TB treatment completed and treatment failure.**

|  | All TB cases | TB completed treatment | Treatment failure |
|---|---|---|---|
|  | n = 90 | n = 55 | n = 35 |
| Patient characteristics |  |  |  |
| Age in years | 45 (18–75) | 45 (18–75) | 44 (20–70) |
| Body Mass Index (BMI) | 17.3 (12.6–31.6) | 17.7 (31.6–13.1) | 16.6 (24.6–12.6) |
| Gender (male) | 75 (83) | 42 (76) | 33 (94) |
| Cough ≥2 weeks | 85 (94) | 52 (95) | 33 (94) |
| Fever | 72 (80) | 44 (80) | 28 (80) |
| Tobacco and/or smoking last 6 months | 51 (57) | 26 (47) | 25 (71) |
| Alcohol past 6 months | 36 (40) | 16 (29) | 20 (57) |
| Previously diagnosed TB[a] | 24 (27) | 12 (22) | 12 (34) |
| Severe thinness[b] |  |  |  |
| BMI < 16.00 kg m$^{-2}$ | 33 (37) | 15 (27) | 18 (51) |

Data shown as counts and percentages except for age and BMI (summarized using mean and range).
[a]Identified as new TB cases by the Revised National Tuberculosis Control Program (RNTCP).
[b]According to the international classification of severe thinness using BMI and the WHO guideline chart.

signatures were similar although no improvement was observed when using combined data (Fig. 2d). In short, it was the same picture as for the baseline signatures using transcriptional data, but no so for the 2-month signatures. The corresponding signature coefficients are reported in Table 4. For a majority of the biomarkers up-regulation implied higher chance of being cured. Only IL7 was shared between the baseline and 2-month signatures.

**Prediction of treatment outcome at 2 months (Early Responders).** The signature for predicting Early Responders vs Late Responders/Non-responders after 2 months ATT using baseline patient characteristics was slightly superior to using baseline transcriptional biomarkers alone (AUC of 0.77 vs 0.67, Table 2) but the combination of both types of data resulted in much better predictions (AUC of 0.97, Fig. 4a). Interestingly, there is not much overlap between transcriptional biomarkers that were

**Table 2 Area under the curve (AUC) values with 95% confidence intervals for baseline and 2-month signatures for prediction of treatment outcome at 2 months (Early Responders) and at 6 months (TB Cures).**

| Validation | Outcome | Time of prediction | N | Patient characteristics | Transcriptional biomarkers | Patient + transcriptional biomarkers | N | Patient characteristics | Protein biomarkers | Patient + protein biomarkers |
|---|---|---|---|---|---|---|---|---|---|---|
| Internal | TB Cures (6 months) | Baseline | 90 | 0.78 (0.68–0.88) | 0.78 (0.69–0.88) | 0.89 (0.83–0.95) | 83 | 0.78 (0.68–0.88) | 0.69 (0.57–0.81) | 0.83 (0.74–0.92) |
|  | TB Cures (6 months) | 2 months | 67 | 0.83 (0.71–0.94) | 0.84 (0.73–0.95) | 0.97 (0.91–1.00) | 59 | 0.74 (0.59–0.89) | 0.68 (0.51–0.85) | 0.68 (0.51–0.84) |
|  | Early responders (2 months) | Baseline | 90 | 0.77 (0.66–0.87) | 0.67 (0.55–0.79) | 0.97 (0.94–1.00) | 83 | 0.74 (0.63–0.85) | 0.78 (0.68–0.88) | 0.88 (0.80–0.95) |
| External | TB Cure (6 months) | Baseline | 84 | 0.71 (0.58–0.85) | 0.74 (0.54–0.94) | 0.92 (0.84–1.00) | – | – | – | – |

predictive f for TB Cures at 6 months and for Early Responders, respectively; only *CXCL10* was predictive for both outcomes (Table 3).

For protein markers the picture was similar although the signatures based on combined data only resulted in a small increase of AUC (Table 2, Fig. 4b). Notably, the biomarkers Eotaxin3, IL7 and MDC were shared between signatures for prediction of Early Responders and for prediction of TB Cures at 6 months. More protein biomarkers were predictive for Early Responders than for TB Cures at 6 months (Table 4), and for most biomarkers up-regulation implied higher chance of being cured.

**External validation of baseline signatures.** Applying the derived baseline signatures for predicting 6-month treatment outcome (using transcriptional biomarkers) in the CTRC cohort resulted in AUC values that were of similar magnitudes as the ones found using internal validation only (Table 2). Moreover, there was the same trend towards improved predictions when combining patient characteristics and biomarker data.

## Discussion

Tuberculosis (TB) patients are highly heterogeneous when it comes to the extent of disease, immune activation and vulnerability to treatment failure. Therefore, tools guiding individual management and treatment are likely to increase cure rates and be more cost-effective and would also largely facilitate the clinical evaluation of new drugs and regimens. Although recent studies have explored the capacity of host biomarker signatures to predict the outcome of TB treatment[9,21,22] none has hitherto taken patient characteristics into account as likely important predictors of TB outcome. Notably, combining patient characteristics with host biomarkers for identification of at-risk subjects has been applied with success both within infectious[23] and non-infectious[24] diseases. To our knowledge, the present study is the first to investigate this approach in the context of TB and demonstrates promising results for prediction of TB treatment outcomes at diagnosis (AUC of 0.89) and excellent results when prediction is based on re-assessment after 2 months ATT (AUC of 0.97). Interestingly, we also identified both transcriptional and proteomic biosignatures at the end of the 2 months intensive phase treatment, a standard time point for evaluation of the ATT response, capable of predicting Early Responders. Notably, small-inducible cytokine B10 (*CXCL10*/IP10) was present in both transcriptional and proteomic signatures and has previously been suggested as a surrogate marker for monitoring sputum culture conversion[25]. These results warrant optimism for tailored treatment for instance by prolonging the induction phase in patients with high risk for treatment failure. The marked differences between baseline and 2-month signatures may be explained by the first 2 months of ATT being an intensive phase with 4 drugs provided (whereas later it becomes a continuation phase with only two drugs) and the intensive treatment may rapidly trigger initial improvements as evident in some biomarkers whereas other biomarkers show up more pronounced during the continuation phase of treatment[26].

For comparison, the 13-gene FAILURE signature reported by Thompson et al.[9] measured at diagnosis that predicted TB treatment failure with an AUC of 0.87, and the 3-gene (*GBP5, DUSP3* and *KLF2*) TB score reported by Warsinske et al.[21] with an AUC of 0.93. Suliman et al.[22] attempted to reduce the number of genes in the 16-gene COR signature[8], paving the way for a simplified point-of-care test, and reported a RISK4 signature (*GAS6, SEPT4, CD1C* and *BLK*) which predicted risk of TB progression on multi-cohort test sets from South Africa, The

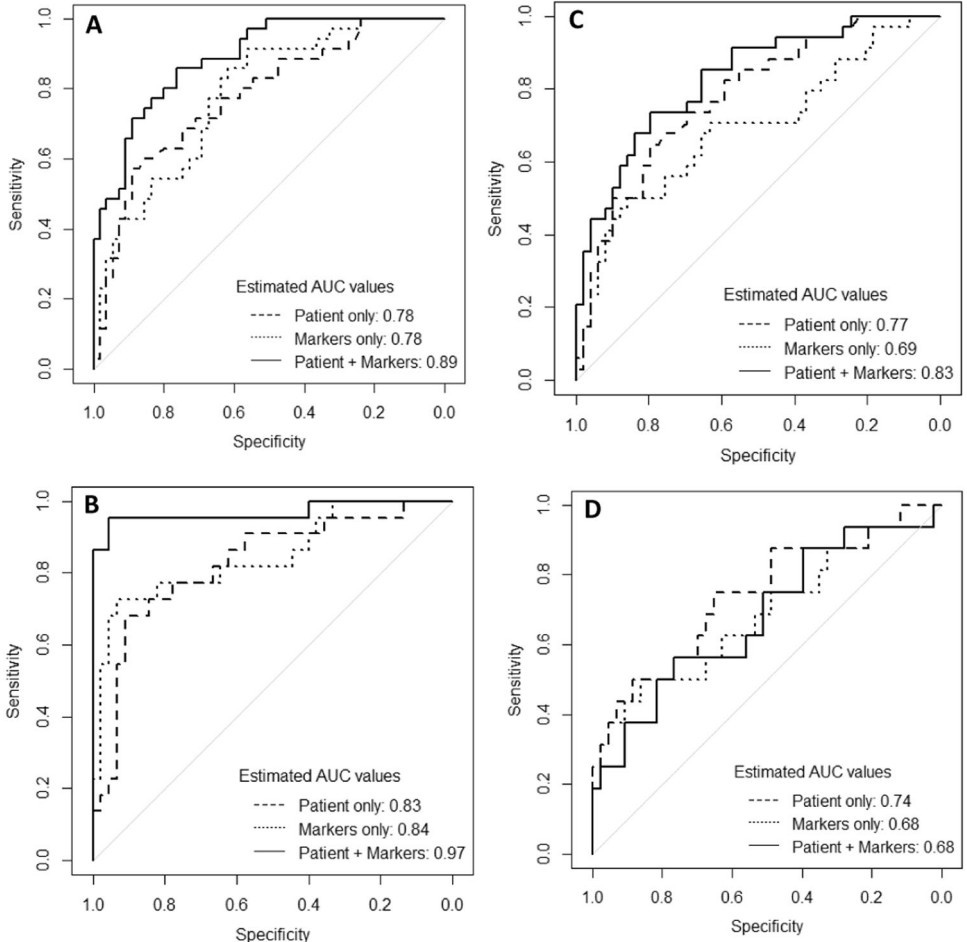

**Fig. 2 ROC curves for signatures predicting treatment outcome at 6 months ATT (TB treatment completed vs. treatment failure). a** Using baseline patient characteristics and/or transcriptional biomarkers. **b** Using 2-month patients characteristics and/or transcriptional biomarkers. **c** Using baseline patient characteristics and/or protein biomarkers. **d** Using 2-month patient characteristics and/or protein biomarkers.

Gambia and Ethiopia (AUCs: 0.67–0.72). Our selected dcRT-MLPA gene panel sets covered only one (*GBP5*) from the 3-genes panel and 6 (*FCGR1A, GBP1, GBP2, GBP5, STAT1* and *TAP1*) out of the 16 COR genes. Thus, it was not possible to make direct comparisons with the outcomes of these studies. But where some overlap between the panels exists, broadly similar results were obtained. Taking into account that the 16-gene COR signature has hitherto been tested only in African populations, the fact that compatible results have been found in an Indian semi-rural population justifies optimism towards the possibility of establishing a universal signature (potentially covering the diversity of *Mtb* lineages present) for treatment response. This suggests that refined signatures that offer even better predictive ability may be identifiable. Further, the present study indicates that three genes (*NLRP1, NOD2* and *TLR6*) measured at baseline appear to be highly predictive of TB treatment completion, suggesting a potential as a minimal signature for monitoring and tailoring ("personalizing") anti-TB therapy.

There are several limitations to the present study. Firstly, signatures were derived using a substudy of a prospective cohort that was not designed for signature discovery. No a priori sample size calculation was carried out as there was no information available on incidence of TB in the study area. Specifically, the recruitment for the cohort was solelyrelying on referral of patients from the RNTCP centres. All eligible patients were enroled in the cohort study. Recruitment relied on pragmatic and selective, but unpredictable sampling scheme of available patients over a

certain time period in a certain area (as is often the case for prospective cohorts). Consequently, the reported results should be seen as exploratory results that should be cautiously interpreted as lack of power could have affected the results obtained in particular in Step 1 of the data analysis (where statistical significance tests were used). Validation in future studies is much needed. To our knowledge this limitation is common to all current TB studies used for signature discovery. Secondly, there were intermittent missing values and drop-outs in the cohort study. The linear mixed models used in step 1 of the data analysis will ensure unbiased results as long as the reasons for missing values and drop-outs can be explained based on the available information about the patients: As far as we can tell there was mostly an explanation such as migration for work, contaminated samples, other technical errors when handling samples, or not getting samples from India to Bergen, Norway (where they were analyzed). Therefore, the linear mixed models most likely produced unbiased estimates. Moreover, it should be noted that the linear mixed models in any case used data from 90 patients, but patients that dropped out contributed less information; in a sense the linear mixed models exploited the available in an optimal way. The present study has also some additional limitations: (a) a low treatment success rate, which could possibly be explained by the fact that ATT was handled in the context of the RNTCP centres, whereas the study team did not have access to the same amount of health information relevant for individualized motivation and support, (b) the inability to detect relapse and (c) the fact that this

**Table 3 Coefficients of baseline and 2-month signatures predicting 6-month treatment outcome (TB cure vs. treatment failure) and baseline signatures predicting 2-month treatment outcome (early responders vs. late responders/non-responders), using patient characteristics and/or transcriptional biomarkers.**

| | 6-month treatment outcome | | | | | | 2-month treatment outcome | | |
| --- | --- | --- | --- | --- | --- | --- | --- | --- | --- |
| Time of prediction | Baseline | | | 2-month | | | Baseline | | |
| Coefficient[a] | Patient | Biomarker | Patient + Biomarker | Patient | Biomarker | Patient + Biomarker | Patient | Biomarker | Patient + Biomarker |
| (Intercept) | 4.3119 | 1.3604 | 4.6141 | 5.0328 | 2.3519 | 15.3840 | 4.9672 | 0.6112 | 8.3258 |
| Age | −0.0028 | | | −0.0041 | | −0.0343 | −0.0186 | | −0.0277 |
| Alcohol past 6 months | −0.4390 | | −0.2522 | −0.3220 | | −0.1609 | −0.6392 | | −1.0338 |
| BMI | | | | −0.1641 | | −0.2430 | | | |
| Gender | −0.9292 | | −0.5899 | −0.3990 | | −0.8762 | −0.3469 | | −0.4981 |
| Cough ≥2 weeks | | | | | | −0.5953 | | | |
| Fever | | | | | | 0.3370 | | | |
| Previously diagnosed TB | −0.4094 | | −0.2527 | −0.2977 | | | −0.8838 | | −1.9152 |
| Severe thinness | −0.8418 | | −0.5385 | −0.2718 | | −0.8243 | | | |
| Tobacco last 6 months | −0.5725 | | −0.3560 | −0.5555 | | −1.6939 | −0.3620 | | −0.1150 |
| ABR | | | | | | | | | −0.1998 |
| AIRE | | −0.8898 | −1.5379 | | | | | | |
| B2M | | 0.0003 | | | | | | | 0.0109 |
| BCL2 | | | | | | −0.0082 | | | |
| CASP8 | | | | | | | | | 0.0323 |
| CCL4 | | | | | | | | −0.0276 | |
| CD14 | | | | | | | | | 0.1130 |
| CD19 | | −0.3668 | −0.4573 | | | | | | |
| CD3E | | | | | −0.0622 | | | | |
| CTLA4 | | | | | | | | | 0.6366 |
| CXCL10 | | | −0.0761 | −1.0138 | | −1.7019 | | | −1.1113 |
| CXCL13 | | −0.0936 | −0.1847 | | | | | | |
| DSE | | | | | | | | | −0.8577 |
| GBP2 | | | | | | | | | −0.0171 |
| GNLY | | | | | | | | | −0.0198 |
| IL13 | | | | | 0.2078 | 0.4596 | | | |
| NCAM1 | | | −0.0368 | | | | | | |
| NLRC4 | | | 0.0925 | | | | | | 1.9794 |
| NLRP1 | | −0.0371 | −0.0816 | | | | | | |
| NLRP2 | | | 0.1218 | | | | | | |
| NOD2 | | −0.3507 | −0.4840 | | | | | | |
| PTPRCv1 | | −0.0043 | | | | | | | |
| RAB13 | | | | | | | | | −0.5528 |
| STAT2 | | | | | | −0.2198 | | | |
| TAP2 | | −0.0712 | | | | | | | |
| TGFB1 | | | | | | | | | −0.0341 |
| TLR4 | | | | | | | | | −0.5711 |
| TLR6 | | −0.1772 | −0.2000 | | | | | | |
| TLR7 | | | | −0.7562 | | −1.3783 | | | 1.1187 |
| TLR8 | | −0.0011 | −0.0150 | −0.0206 | | −0.0001 | | | |
| TNIP1 | | | | | | | | | −0.5144 |

[a]Estimated coefficients of biomarkers have been multiplied by 1000 for ease of representation; see Appendix 1 for an explanation on how to use the coefficients.

study surveyed only a limited number of genes. Moreover, in this study a dichotomized outcome (completed vs. failure) was used, potentially ignoring some information, such presence or absence of cavitations on chest X-rays, which could be indicative of differences that would also be seen in signatures. Although RNA sequencing offers the advantage of being an unbiased approach compared to pre-selected genes assessed with the dcRT-MLPA method, dcRT-MLPA was chosen as it offers practical advantages for clinical studies, in particular, its robustness, user-friendly approach and cost-effectiveness (where dcRT-MLPA is also compared to RNA-Seq[10]).

Our findings emphasise the importance of combining host-derived biomarkers with basic patient characteristics in predicting the TB treatment outcomes. Based on these results, further refinement and validation of signatures of TB treatment response are warranted to enable their use to guide individually tailored TB therapy and as read-outs in efficacy studies of new treatment regimens and strategies. As a longer-term goal, we plan to compare gene expression signatures identified in this study with those from household TB contacts to see if we can identify genes generally involved in *Mtb* control (non-progression in *Mtb*-infected patients) and disease resolution.

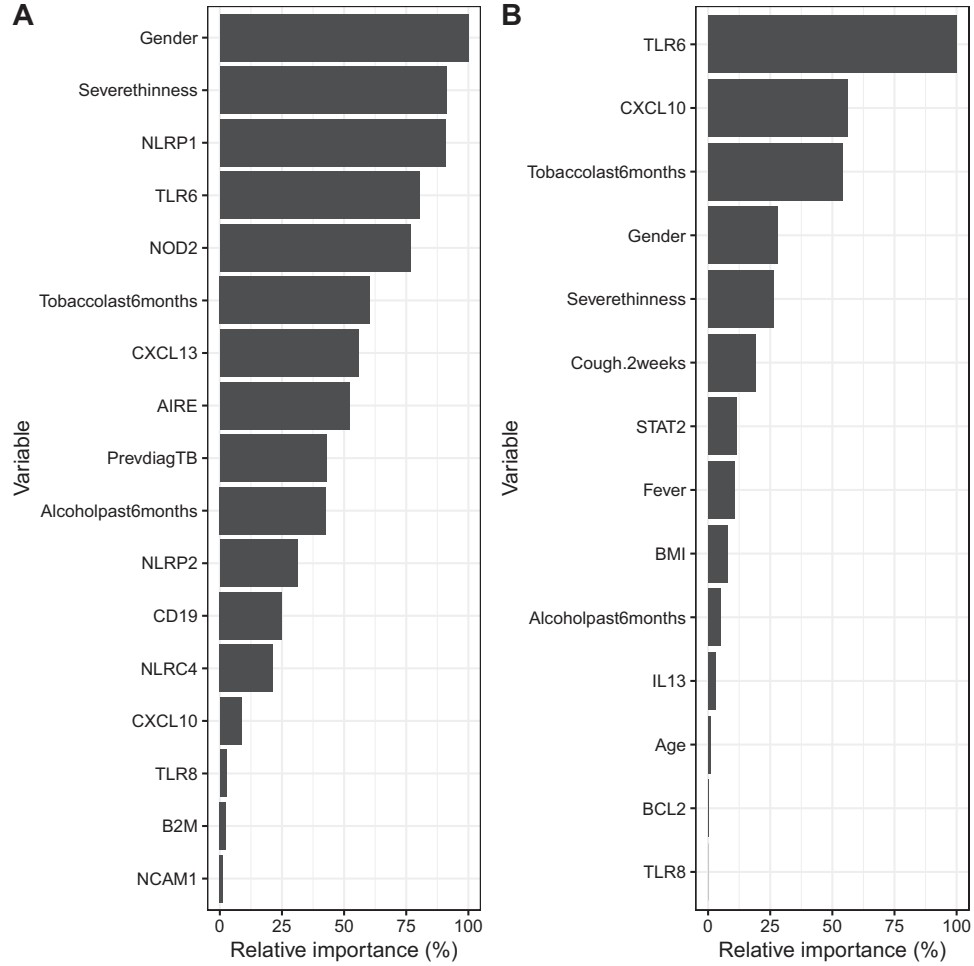

**Fig. 3 Bar plots showing the relative importance of patient characteristics and transcriptional biomarkers (combined data). a** Signature for predicting treatment outcome at 6 months ATT using combined baseline data. **b** Signature for predicting treatment outcome at 6 months ATT using combined 2-month data.

## Methods

**Ethical consideration**. Ethical approval for this study was obtained from the Institutional Ethical Review Board (IERB) of St. John's Medical College, Bangalore (IERB/1/527/08, July 15 2008). The material transfer agreement between St. John's Medical College, Bangalore and the University of Bergen, Norway was obtained from the Department of Biotechnology, Government of India (No.BT/Med.II/Adv (SS)/Misc./02/2012). Ethical approval was also obtained (Ref no: 2018/1614 D) from the Regional Committee for Medical and Health Research Ethics, Western Norway.

**Study population**. This study is a substudy nested within a prospective cohort study of adult PTB index cases and their household contacts was conducted in Palamaner and Kuppam Taluks, Chittoor district, Andhra Pradesh, India, between September 2010 and April 2012, to measure the incidence of *Mtb* infection and TB disease in a highly-exposed population. Participants were enroled upon referral from the Revised National Tuberculosis Control Program (RNTCP) centres of the Chittoor district. The present substudy only included the index PTB cases that provided blood samples at baseline. Recruited PTB index cases were treated with standard ATT and followed during their 6-month ATT course.

**External validation data**. Data from a well-characterized TB treatment cohort, the Catalysis treatment response cohort (CTRC), were used for external validation. The CTRC consists of HIV-negative patients from South Africa, diagnosed with pulmonary TB. WB transcriptomes were measured via RNA sequencing[9,27]. Patients with culture-confirmed TB were assessed mainly based on sputum culture status after 6 months of standard ATT. Previously, this cohort has been used to identify transcriptional signature that predict treatment failure at baseline. For external validation baseline transcriptional biomarker data and 6-month treatment outcome data were used. There was, however, no information on fever and alcohol consumption and no RNA sequencing data on the gene *CXCL13* (missing due to sequencing depth covered). There were also no protein data available.

**Clinical assessments and sampling**. Baseline assessments: medical history (including BCG vaccination status, history of TB exposure, prior TB and habitual risk factors), demographic, anthropometric and clinical data were recorded. At baseline, a tuberculin skin test (TST) was performed by a trained nurse (2 TU/0.1 mL tuberculin; Span Diagnostics, Surat, India) and read after 48–72 h; an induration ≥10 mm was defined as positive. The chest X-ray (anteroposterior view) at baseline was interpreted by three independent radiologists. Agreement by at least two radiologists was required for the radiological diagnosis of PTB (discordant readings were discarded). Patients were tested for HIV after pre-test counselling although agreement for HIV testing was not a pre-requisite for participation.

Longitudinal sampling: sputum samples were collected from all PTB index cases at 0, 1, 2 and 5 months; (two samples on two consecutive days at baseline, one sample at the remaining time points) for fluorescent microscopy (auramine) and culture (Mycobacterial Growth Indicator Tube, BD) as described previously[28]. Positive cultures were confirmed by using the HAIN kit (GenoType MTBC, Hain Lifescience GmbH, Nehren, Germany). WB for biomarker analysis was collected at 0, 1, 2 and 6 months. Peripheral blood (3 ml) was drawn for the QuantiFERON®—TB Gold In-Tube (QFT-GIT) test (Cellestis, Australia) at 0, 1, 2 and 6 months. Body Mass Index (BMI; weight (kg)/height² (m)) was evaluated at 0, 1, 2 and 6 months of treatment follow-up.

**Definition of treatment completed**. We used this definition: "Patients with TB who completed treatment without evidence of failure but with no record to show that sputum smear or culture results in the last month of treatment and on at least one previous occasion were negative, either because tests were not done or because results are unavailable"[18]. Sputum was sampled after 5 months of ATT. Culture negative cases at this time point were designated "TB treatment completed". Cases who failed to achieve bacteriological cure at 5 months were designated "Treatment failure" in accordance with WHO guidelines[18]. Moreover, the additional sampling of longitudinal sputa in the present study allowed us to define more treatment outcomes based on sputum culture status at 2 months. i.e., (a) Early Responders—*Mtb* culture negative at 2 and 5 months, (b) Late Responders—*Mtb* culture positive

**Table 4 Coefficients of baseline and 2-month signatures predicting 6-month treatment outcome (TB cure vs. treatment failure) and baseline signatures predicting 2-month treatment outcome (early responders vs. late responders/non-responders), using patient characteristics and/or protein biomarkers.**

| Time of prediction | 6-month treatment outcome | | | | | | 2-month treatment outcome | | |
| --- | --- | --- | --- | --- | --- | --- | --- | --- | --- |
| | Baseline | | | 2-month | | | Baseline | | |
| Coefficient[a] | Patient | Biomarker | Patient + Biomarker | Patient | Biomarker | Patient + Biomarker | Patient | Biomarker | Patient + Biomarker |
| (Intercept) | 4.5593 | −0.9129 | 2.9571 | 0.4569 | −1.2491 | −1.0408 | 3.0551 | 0.0177 | 4.7033 |
| Age | −0.0070 | | | | | | −0.0129 | | −0.0192 |
| Alcohol past 6 months | −0.2634 | | −0.4249 | | | | −0.4082 | | −0.6797 |
| BMI | | | | −0.0401 | | | | | |
| Gender | −0.9697 | | −0.5714 | | | | −0.1290 | | −0.3000 |
| Cough ≥2 weeks | | | | | | | | | |
| Fever | | | | | | | | | |
| Previously diagnosed TB | −0.3919 | | −0.4778 | | | | −0.3565 | | −1.1841 |
| Severe thinness | −0.9332 | | −0.7195 | −0.2969 | | | | | |
| Tobacco last 6 −months | −0.6471 | | −0.3924 | −0.1608 | | | −0.2935 | | −0.1728 |
| Eotaxin3 | | | 0.2167 | | | | | 0.8981 | 1.3535 |
| GMCSF | | 2.0843 | 3.8132 | | | | | | |
| IL1b | | | | | | | | 0.0049 | 0.2252 |
| IL4 | | | | | | | | 538.5312 | 84.1561 |
| IL7 | | 44.7181 | 40.6383 | | 15.7258 | 3.6989 | | 31.6772 | 16.3365 |
| IP10 | | | | | | | | 0.0028 | 0.0026 |
| MDC | | −0.1552 | −0.2075 | | | | | −0.5229 | −0.5512 |
| VEGF | | | | | | 0.1529 | | 0.8861 | 1.3231 |

[a]Estimated coefficients of biomarkers have been multiplied by 1000 for ease of representation; see Appendix 1 for an explanation on how to use the coefficients.

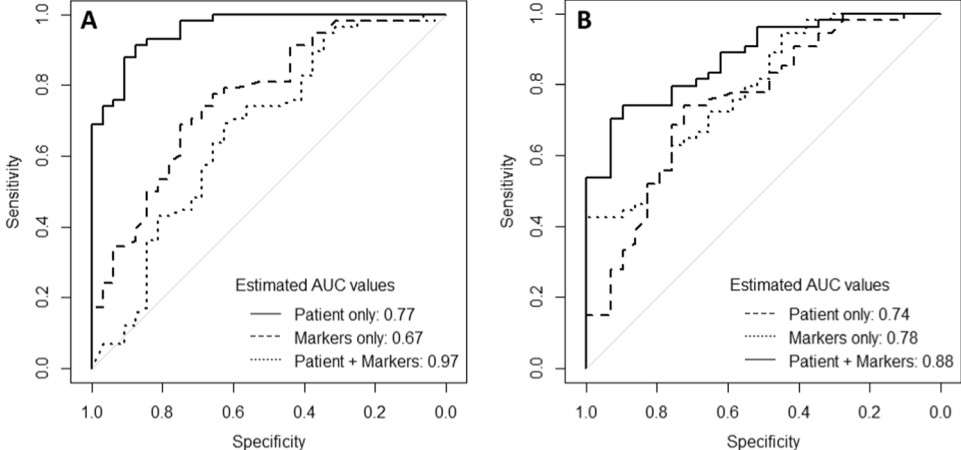

**Fig. 4 ROC curves for signatures predicting treatment outcome at 2 months ATT (early responders vs. late responders/non-responders). a** Using baseline patient characteristics and/or transcriptional biomarkers. **b** Using baseline patients characteristics and/or protein biomarkers.

at 2 months but *Mtb* culture negative at 5 months and c) Non-responders/Treatment Failures—*Mtb* culture positive at 5 months.

**Sample collection and RNA extraction**. Peripheral WB (~2.5 ml) was drawn into PAXgene blood RNA tubes (PreAnalytiX, Hombrechtikon, Switzerland) and stored at −80°C until RNA extraction (PAXgene Blood RNA kit; PreAnalytiX, Hilden, Germany). Total RNA concentration and purity were measured using a Nanodrop spectrophotometer (Thermoscientific, Wilmington, DE, USA) and ranged between 0.4 and 13.2 μg (average ± SD: 3.8 ± 1.65 μg).

**Selection of transcriptional biomarkers**. A total of 198 genes (including four housekeeping genes), distributed in three panels were assessed; the first 48-gene set (identified by the partners in the Bill and Melinda Gates Foundation Grand Challenge

project #6 consortium) has been described in our previous studies[11–14]. The second 92-gene set included genes known to be involved in general inflammation and myeloid cell activation, and genes involved in the adaptive immune system, comprising Th1/Th2-responses, regulatory T-cell markers and B-cell associated genes. The third 58-gene set included type 1-interferon-inducible genes known to be up-regulated in adult TB and genes associated with prediction of TB risk in South African neonates. In total, thirty genes were present in more than one panel. There were in total 145 unique genes (see Supplementary Data). For the 30 repeated genes that were present in more than one panel geometric mean expression was used.

**Dual-colour-Reverse-Transcriptase-Multiplex-Ligation-dependent-Probe-Amplification (dcRT-MLPA)**. For each target sequence, a specific RT primer was designed, located immediately downstream of the left- and right-hand half-probe

target sequence. A total of 125 ng RNA was used for reverse transcription, applying MMLV reverse transcriptase (Promega, Madison, WI, USA), followed by hybridization of left- and right-hand half-probes to the cDNA at 60° C overnight. The remaining steps were performed as described elsewhere[14]. All 268 samples were run in three (96-well) plates for each of the gene panels. The PCR fragments were analysed on a 3730-capillary sequencer in Gene scan mode (Life Technologies, Carlsbad, CA, USA), using GeneMapper version 5.0 (Life Technologies, Carlsbad, California, USA). Primers and probes were obtained from the Department of Infectious Diseases, Leiden Medical University, the Netherlands. GAPDH was used for normalization.

**Multiplex cytokine/chemokine assays.** Biomarkers at the proteomic level were analysed in peripheral WB stimulated with a mixture of *Mtb*-specific antigens (Early Secretory Antigenic Target-6 (ESAT-6), Culture Filtrate Protein-10 (CFP-10) and TB antigen 7.7—QFT-GIT supernatants). Six of ten biomarkers from each panel (pro-inflammatory panel—IL1b, IL10, IL4, IL12p70, IFNγ and TNFα; cytokine panel—GM-CSF, IL15, IL17A, IL5, IL7 and VEGF; chemokine panel—Eotaxin3, IL8, IP10, MCP1, MDC and MIP1b) were analysed.

**Data analysis.** Patient characteristics were summarized using mean and range or count and percentage, as appropriate. Cases of TB treatment completed and treatment failures were compared using the Mann-Whitney test, Pearson's chi-square test with Yates Continuity Correction, or Fisher's exact test, as appropriate.

Signatures were identified using a two-step approach; such approaches are commonly used for signature identification[9]. In the first step, univariate filtering (marker-by-marker feature selection) was applied to identify and shortlist biomarkers that were potentially predictive; this step may be seen as dimension reduction, omitting redundant biomarkers. In the second stage, multivariate modelling reduced the shortlisted markers to markers that were not too correlated and still predictive. Univariate filtering was carried out by means of longitudinal data analysis where biomarker profiles over time were estimated and compared between groups (completed vs. failed) by means of appropriate models for repeated measurements over time[9]. In this step, biomarkers were shortlisted based on statistically significant differences between groups at baseline or later time points as appropriate. In the second step, an inverse regression approach was used such that group membership (completed vs. failed) became the binary response in a LASSO (Least Absolute Shrinkage and Selection Operator) logistic regression model[29]. All shortlisted biomarkers were included simultaneously as explanatory variables in the LASSO model; this approach provided an effective means to handle both correlation and multiplicity issues[30]. The two steps are described in detail below.

Step 1: Longitudinal data analysis: longitudinal changes in biomarker expression levels after 1, 2 and 6 months of ATT for all index PTB patients and within the two treatment outcome groups were assessed by means of linear mixed models. Measurements below the lower detection limit were accommodated as left-censored observations in the linear mixed models[31]. Models were fitted to base 2 logarithm transformed measurements. Group-time interactions were included as fixed effects. Patient-specific random effects were also included. Differences in mean levels were evaluated based on post hoc t-tests derived from the linear mixed models. All biomarkers that showed statistically significant differences between groups at baseline, 2 months, or 6 months were included for signature identification (step 2 described below). Note that 1 month data were included in the linear mixed models to fully exploit the available data even though no identification of signatures at visit 1 would be possible. In short, step 1 aimed at modelling biomarker levels in such a way that important features in the data were incorporated (left censoring and unbalanced repeated measurements).

Step 2: Signature identification: the reverse regression approach implied that group membership became the outcome and biomarker levels became the exposure in step 2, which was the opposite of the modelling approach used in step 1. Specifically, modelling was carried out using LASSO regression, which attempts to assign the value 0 to slope coefficients of redundant markers, i.e., markers that are highly correlated to other markers retained in the model. This is achieved by using a penalty term in the estimation procedure: the penalty ensures that similar coefficients will be forced to become equal to 0 to achieve the optimal model fit[29]. The penalty term has a tuning parameter, usually referred to as lambda, which controls how similar biomarkers should be before one of them receives a slope coefficient of 0. The optimal lambda is determined based on the data by means of cross-validation where 90% of the data is used to fit ("train") the model and the remaining part used to benchmark or validate through comparison of actually observed data and predictions from the trained model. The optimal lambda is the lambda values leading to the smallest prediction error. Ten-fold cross-validation carries out such comparisons ten times based on a random partitioning of the data into ten parts of equal size. Optimal LASSO models correspond to optimal lambda values. However, as ten-fold cross-validation relies on random sampling, it may not necessarily produce the exact same results when being repeated. Therefore, to stabilize results, the ten-fold cross-validation step was repeated 100 times. The use of cross-validation serves as internal validation of the signatures.

All shortlisted markers from step 1 were included simultaneously in the LASSO regression models, with or without additionally including the basic patient characteristics identified in the literature [age, gender, cough (last 2 weeks), fever, previously diagnosed TB, tobacco/smoking (last 6 months), alcohol consumption (last 6 months), BMI and severe thinness (also called severely low BMI, a binary

indicator of BMI < 16.0 kg/m³⁰)][32]. A LASSO regression model only including basic patient characteristics was also fitted. Results were reported as estimated intercept and slope coefficients for the identified patient characteristics and biomarkers. Receiver operating characteristic (ROC) curves were also shown and summarized using the area under the curve (AUC). The relative importance of identified patient characteristics and biomarkers was evaluated in terms of percentages reflecting magnitudes of estimated coefficients relative to the maximum coefficient among identified patient characteristics and biomarkers (in absolute terms, ignoring signs of coefficients); important variables would have percentages above 25%[33].

For transcriptional biomarkers, step 1 and 2 were applied to identify baseline and 2-month signatures predicting treatment outcome at 6 months (TB treatment completed vs. treatment failure). Likewise, for transcriptional biomarkers, step 1 and 2 were also applied to find baseline signatures predicting the treatment outcome at 2 months (Early responders vs. Late responders/Non-responders). For protein biomarkers only step 2 was applied as there was no need for univariate filtering due to the small number of available biomarkers.

External validation: The identified transcriptional signatures were evaluated on the RNA sequencing data (Gene Expression Omnibus: GSE89403)[9]. No suitable data with clinical outcomes of TB treatment and follow-up samples were available for the proteomic signature evaluation. In contrast to our study, the validation data did only provide basic patient characteristics at baseline. The entire identified signatures (including slope coefficients) were not applicable to the external validation data as the use of a different platform rendered biomarker levels that had a different variation. Instead, the predictive capability of the signatures was assessed through the fitting of logistic regression models, which included all basic patient characteristics and biomarkers of the identified baseline and 6-month signatures, to the external validation baseline and 6-month data, respectively. AUC values were reported.

Analyses were carried out using an **R** Markdown script (R Core Team, 2019)[34], exploiting the **R** extension packages *glmnet, ggplot2, ggpubr, multcomp, pROC, survival* and *xtable*, through the user interface RStudio (http://www.rstudio.com). The script is available from the autors upon request. A significance level of 0.05 was used in step 1.

**Understanding how the signatures are constructed.** The estimated coefficients may be used to form a linear predictor, which upon back-transformation from the log-odds scale to the probability scale (using the inverse logit transformation) provides the probability of treatment success.

For instance, for the baseline signature derived using both patient characteristics and transcriptional biomarker data, the equation looks like this:

log odds of treatment success
= 4.61 − 0.59 (if female)
− 0.25 (only if previously diagnosed with TB: yes)
− 0.36 (only if tobacco/smoking in the last 6 months: yes)
− 0.25 (only if alcohol consumption in the last 6 months: yes)
− 0.54 (only if severe thinness: yes)
− 0.0015 AIRE level
+ 0.00000027 · B2M level
− 0.00046 · CD19 level
− 0.000076 · CXCL10
− 0.00018 · CXCL13
− 0.000037 · NCAM1
+ 0.000093 · NLRC4
− 0.000082 · NLRP1
+ 0.00012 · NLRP2
− 0.00048 · NOD2
− 0.00020 · TLR6
− 0.000015 · TLR8

The biomarker expression levels to be inserted in the above equation should be the untransformed expression levels. Once the log odds has been calculated the probability of treatment success is obtained as follows:

$$\text{probability} = \exp(\log \text{odds})/(1 + \exp(\log \text{odds}))$$

The above coefficients of the biomarkers look small, but it is because expression levels take on large values. Moreover, it is noteworthy that most coefficients of the signature used in the above example are negative, implying that the larger the expression level the less likely is treatment success, i.e., up-regulation means less chance of treatment success.

**Reporting summary.** Further information on research design is available in the Nature Research Reporting Summary linked to this article.

## Data availability
All relevant data are available upon request. Please contact the corresponding author. Source data underlying plots shown in figures are provided in Supplementary Data 2.

## Code availability
The custom R code used is deposited on Zenodo.org (https://doi.org/10.5281/zenodo.3899753).

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

## Acknowledgements

We thank Drs Rajini Macaden, Nelson Jesuraj, Anto Jesuraj Udaykumar and Vandana A at St. John's Research Institute, Bangalore; Aud Eliassen at the sequencing laboratory at Haukeland University Hospital, Bergen, Norway; Gry Klouman Bekken at Department of Infection control, University of Oslo for inputs towards organisation of the data set for the household TB contact study. We thank Aeras (a non-profit organisation), USA for their contributions to establishing the TB vaccine trial site at Palamaner Taluk, Chittoor district, Andhra Pradesh, India; Meso Scale Discovery (MSD), USA for the loan of the instrument and for the technical support and advise provided by Tynde Sandor (MSD) and Gail Calvert (MSD). Research Council of Norway Global Health and Vaccination Research (GLOBVAC) projects: RCN 179342, 192534 and 248042, the University of Bergen (Norway); EDCTP2 programme supported by the European Union; the St. John's Research Institute, Bangalore. We also acknowledge EC FP7 ADITEC (Grant Agreement No. 280873); EC HORIZON2020 TBVAC2020 (Grant Agreement No. 643381) [the text represents the authors' views and does not necessarily represent a position of the Commission who will not be liable for the use made of such information].

## Author contributions

D.S., S.J., M.V., T.M.D., C.R. and H.M.S.G. conceptualized and designed the biomarker study. S.S. and M.V. coordinated patient recruitment and follow-up. D.S., S.J. and C.R. wrote the manuscript with contribution from M.V., T.M.D. and H.M.S.G. D.S. performed all laboratory experiments. C.R. and D.S. performed the data analysis and generated tables and figures. T.H.M.O. and M.C.H. contributed to reagents and protocols for the dc-RT MLPA and reviewed the manuscript. S.T.M. provided CTRC cohort information and reviewed the manuscript. C.R. supervised the statistical analysis, wrote the section on statistical analysis and reviewed the manuscript. C.R. and H.M.S.G. had primary responsibility for the final content of the manuscript. All authors have approved and contributed to the final manuscript.

## Competing interests

The authors declare no competing interests.
