## [Peer Review File · Communications Biology]

Reviewers' comments:

Reviewer #1 (Remarks to the Author):

The authors have performed an elegant study to define host biomarkers and clinical characteristics that predict treatment failure in a small sample of pulmonary TB patients. The findings are novel and interesting but the authors need to address the following concerns.

1. The abstract is a bit scattered and unclear. Kindly summarize the details in a bit more coherent manner.
2. The methods does not mention how sample size calculations were done. How were the patients screened? Were all TB patients recruited or was there selective sampling? What were the exclusion criteria?
3. Is the term severe thinness accepted terminology? Would severely low BMI be a better terminology?
4. What was the drug resistance profile of these TB patients? Was DR TB excluded? If not, how do the authors account for the fact that DR TB could impact baseline or 2 month signatures?
5. Very little description of the external validation cohort is available. Please expand on this? Is there a validation cohort for the protein signatures?
6. The treatment failure rate of 38.9% is inordinately high for Indian patients. The RNTCP has consistently reported higher treatment success rates and this has been replicated in several clinical trials across India. The authors of the study need to perform a more serious exercise to decipher the cause of this high failure rate. Lack of adherence would confound their biosignatures and nullify their efforts.
7. The results section should clearly mention the list of host biomarkers and clinical characteristics used as predictors and not just refer to the Table.
8. What is the reason for the differences in the biosignatures at 0 and 2 months?
9. It is unclear to this reviewer as to why we need predictors of 2 month outcomes (results + figures). It seems to me no treatment decision alterations will be made if the patient is a slow or fast responder as long as he/she responds at 5/6 months.
10. The Methods section should elaborate on how the protein signature is measured. Was plasma/serum used? Was it only the QFT supernatants? Currently it appears that the proteins were measured in the TB antigen stimulated QFT supernatants? What happens in the nil antigen tube?

Reviewer #2 (Remarks to the Author):

Thank you for submitting this very interesting paper. It is quite exciting to see how patient characteristics and biomarkers can contribute together to predict outcomes.

My main concerns are with the sample size/power calculation, which I think should be included, and the number of participants who were lost to follow-up (the 5 month follow-up only included 41% of the original participants). This might have introduced bias which should be addressed in the limitations section.

In addition, I have added a few comments in the manuscript attached to this email.

Response to Reviewers

Reviewers' comments:

Reviewer #1 (Remarks to the Author):

The authors have performed an elegant study to define host biomarkers and clinical characteristics that predict treatment failure in a small sample of pulmonary TB patients. The findings are novel and interesting but the authors need to address the following concerns.

Response: *Thank you very much for the appreciation of our work. Below we've responded to your concerns.*

1. The abstract is a bit scattered and unclear. Kindly summarize the details in a bit more coherent manner.

Response: *The abstract has now been substantially shortened to 148 words.*

2. The methods does not mention how sample size calculations were done. How were the patients screened? Were all TB patients recruited or was there selective sampling? What were the exclusion criteria?

Response: *Thanks for making this point. The present study is based on a cohort study where patients were enrolled upon referral from the Revised National Tuberculosis Control Program (RNTCP) centres of the Chittoor district, if presence of Mtb was confirmed by sputum smear and/or culture and chest X-ray. You may view this as a kind of selective sampling of available patients over a certain time period as is often the case for prospective cohorts. We've provided more details on this process as a limitation in the Discussion.*

3. Is the term severe thinness accepted terminology? Would severely low BMI be a better terminology?

Response: *We believe that severe thinness is accepted terminology, please see here:*

Global Nutrition Report. Shining a light to spur action on nutrition. (2018).

https://reliefweb.int/sites/reliefweb.int/files/resources/2018_Global_Nutrition_Report.pdf

(last assessed:08.04.2020)

However, we've used both terms the first time they occurred in the manuscript.

4. What was the drug resistance profile of these TB patients? Was DR TB excluded? If not, how do the authors account for the fact that DR TB could impact baseline or 2 month signatures?

Response: *With regard to the drug resistance profile in this study, around 18.0% (17/90) of TB cases were identified as drug resistant (DR) TB (resistance to isoniazid) and 2.2% (2/90) of TB cases were identified as MDR TB (resistance to both rifampin and isoniazid). Notably, 20.0% (11/55) and 17.1% (6/35) of DR TB cases distributed more or less equally in the treatment cured and failures, respectively. Thus, we haven't excluded DR TB cases from the biomarker analysis.*

5. Very little description of the external validation cohort is available. Please expand on this? Is there a validation cohort for the protein signatures?

Response: Thank you for pointing this out. We've provided more information. Unfortunately, there is no relevant protein dataset (which uses treatment outcome to identify biomarkers/biosignatures) available for validating the protein signatures.

6. The treatment failure rate of 38.9% is inordinately high for Indian patients. The RNTCP has consistently reported higher treatment success rates and this has been replicated in several clinical trials across India. The authors of the study need to perform a more serious exercise to decipher the cause of this high failure rate. Lack of adherence would confound their biosignatures and nullify their efforts.

Response: We agree with the reviewer that the present study shows high TB treatment failure. The reason for this high treatment failure rate may possibly be due that patients with a history of prior TB were also included in this study. We mention this shortcoming in the Discussion section.

Notably, around 27.0% (24/90) were diagnosed with TB previously, of which half of the patients reported recurrence of TB in < 2 years. Around 29.0% (10/35) of treatment failures had a history of prior TB. However, for the biomarker analysis, we noticed that the presence of a history of prior TB was equally distributed among the TB cures and treatment failures. We mention this shortcoming in the Discussion section.

7. The results section should clearly mention the list of host biomarkers and clinical characteristics used as predictors and not just refer to the Table.

Response: Thanks for pointing this out. We've now listed biomarkers and clinical characteristics for the key results in the text in the Results section.

8. What is the reason for the differences in the biosignatures at 0 and 2 months?

Response: This is a good question. We can only speculate but the first 2 months of ATT is an intensive phase with 4 drugs provided (whereas later it becomes a continuation phase with only two drugs) and the intensive treatment may rapidly trigger initial improvements as evident in some biomarkers whereas other biomarkers show up more pronounced during the continuation phase of treatment. We've included this potential explanation in the Discussion section.

9. It is unclear to this reviewer as to why we need predictors of 2 month outcomes (results + figures). It seems to me no treatment decision alterations will be made if the patient is a slow or fast responder as long as he/she responds at 5/6 months.

Response: We agreed that these results may not be very useful. However, we would prefer to retain these results to offer a more complete picture in terms of changes in biomarkers. These results don't any longer feature prominently in the Discussion.

10. The Methods section should elaborate on how the protein signature is measured. Was plasma/serum used? Was it only the QFT supernatants? Currently it appears that the proteins were measured in the TB antigen stimulated QFT supernatants? What happens in the nil antigen tube?

Response: We agree that this information was missing. Now we have included both transcriptional and protein biomarkers details in the method section. Yes, we used QFT supernatants and measured cytokine/chemokines in the TB antigen stimulated QFT supernatant tubes only, and we didn't use Nil antigen tubes for the biomarker analysis.

Reviewer #2 (Remarks to the Author):

Thank you for submitting this very interesting paper. It is quite exciting to see how patient characteristics and biomarkers can contribute together to predict outcomes.

Response: *Thank you very much for the positive feedback. Below we've responded to your comments and suggestions.*

My main concerns are with the sample size/power calculation, which I think should be included, and the number of participants who were lost to follow-up (the 5 month follow-up only included 41% of the original participants). This might have introduced bias which should be addressed in the limitations section.

Response: *Thank you for these 2 important points.*

The present study is nested within a cohort study where patients were enrolled upon referral from the Revised National Tuberculosis Control Program (RNTCP) centres of the district, if presence of Mtb was confirmed by sputum smear and/or culture and chest X-ray. You may view this is a kind of pragmatic or selective sampling of available patients over a certain time period as is often the case for prospective cohorts. We've clarified the study design in the Methods section. However, it also means that we aren't able to provide a sample size calculation. Furthermore, we've now addressed this as a limitation in the Discussion section.

Moreover, you're right that there is some drop-out over time. If we can entertain the assumption of "missing completely at random" or "missing at random" (missing data were either entirely unrelated to being in the study or they could be explained by the observed data) then the applied linear mixed models (used in the first step) will still provide unbiased results. We believe that in most cases the patients who dropped out did so for reasons unrelated to the study (as detailed in the Discussion).

In addition, I have added a few comments in the manuscript attached to this email.

Response: *Thank you so much for these helpful comments and suggestions, which we've mostly followed as evident in the revised manuscript. Below, we briefly comment on the most important ones:*

- *The introduction has been shortened.*
- *There is no expiry data for the Ethical Approval; the original date is still valid.*
- *The WHO definition of treatment completed has been included.*
- *We've provided more details on the external validation data.*
- *Details on sequencing methodology have been included.*
- *Patient characteristics were identified from the literature. This is now indicated in the manuscript.*
- *It could be that patients with cavitations on CXR could have a different profile (and signature) compared to patients without cavitations. This is an interesting idea. However, we feel it goes beyond the scope of the present manuscript where a dichotomized outcome was used. We've added a sentence in the Discussion.*
- *Good points about the drop-outs, which we failed to address in the manuscript. It should be noted that in our STEP 1 the linear mixed models in any case used data from 90 patients, but then the patients staying in the cohort study (not dropping out) contributed more information although due to the correlation between repeated measurements on the same patients the*

extra information gained may not be that large; this was the very reason for using linear mixed models. This is now mentioned in the Discussion.

- Table 1 and Figure 1 (now in a CONSORT like format) have been revised entirely so that they fit with the present study design, which is a substudy nested within a cohort.

REVIEWERS' COMMENTS:

Reviewer #1 (Remarks to the Author):

I do not have any new comments as most of my concerns have been addressed.

Reviewer #2 (Remarks to the Author):

Thank you for addressing most of the previous points raised.

My remaining concern is with the definition of cured vs. treatment completed (these two together are usually defined as "treated successfully") and all the other possible treatment outcomes are defined as "treated unsuccessfully" or treatment failure. Could the authors make it very clear that they used either "cured", or "completed" or both, vs. treatment failure. At the moment the definition for "completed" is included, but it seems from the text "cured" was rather used? Please clarify throughout the manuscript.